# The Influence of Governmental Agricultural R&D Expenditure on Farmers' Income—Disparities between EU Member States

**Mirela Stoian, Raluca Andreea Ion, Vlad Constantin Turcea \*, Ionut Catalin Nica and Catalin Gheorghe Zemeleaga**

The Department of Agrifood and Environmental Economics, Faculty of Agrifood and Environmental Economics, The Bucharest University of Economic Studies, 010961 Bucharest, Romania
\* Correspondence: turceavlad14@stud.ase.ro; Tel.: +40-72-295-9353

**Abstract:** This article investigates how governmental agricultural R&D expenditure affect economic prosperity and sustainable development, attempting to verify the hypothesis that agricultural research and development expenditures are among the key factors influencing the farmers' income, as one of the sustainable development indicators. Statistical data were retrieved from European international databases for the period of 2004–2020 and were analyzed using the regression model. The results of the study indicate positive effects for most of the EU member states. The countries where the results validate the hypothesis are Austria, Belgium, Bulgaria, the Czech Republic, Germany, Estonia, Finland, France, Greece, Croatia, Ireland, Latvia, Poland, Slovakia, Slovenia, and the United Kingdom, as a former member state of the EU. Further, the model confirms that a significant portion of farmers' income growth is explained by the governmental R&D expenditure. These findings may change the methods and directions regarding the agricultural R&D expenditure, underpinning the macroeconomic policy and agriculture in rural areas along the pathway to achieving the sustainable development goals.

**Keywords:** sustainable agriculture; governmental agricultural R&D expenditure; farmers' income; sustainable development goals

## 1. Introduction

The sustainable development of agriculture is essential for the economic prosperity of the European Union's rural communities. Statistical data show that rural areas represented 83% of the total EU area, and that agricultural land, forest, and natural areas represented 80% of the total EU area, in 2018, as stated by Eurostat. Representative percentages explain the role of the rural communities in the economy, and discussions about macroeconomic policy may focus on a sectors' prioritization, when speaking about funding, investment, and public expenditure and their implementation effectiveness. When investments contribute to the economic prosperity of the rural communities, the macroeconomic policy is considered to be effective. Economic prosperity, in turn, may be described by numerous indicators; among these, the farmers' income can illustrate the economy's current status. A deeper analysis, which explores this indicator and its drivers, is needed in order to visualize the sustainable development of rural areas. Our study starts from the assumption that research and development (R&D) investments in agriculture are among the key factors influencing the levels of farmers' income.

Sustainable agriculture represents the equilibrium point of several aspects, including social, economic, and environmental, in both rural and urban agri-based contexts. For the agriculture to be designated as "sustainable", it is mandatory to have a versatile position, to offer an easy scalability, and to be continuously adaptable [1].

Agricultural industry is also becoming more data-centric, and novel technologies are offering advantages to worldwide farmers. Several break-through sustainable agricultural practices have been highlighted in the research literature, and arguments have been offered for consideration [2].

Many researchers have emphasized the importance of R&D expenditure in agriculture [3–8], arguing a direct and causal relationship between R&D investment in agriculture and farmers' income. However, most of these studies are based on a theoretical and conceptual framework, and only a few of them have conducted an empirical analysis for the EU member states, divided into old and new, with a more or less important agricultural sector. Our study explores the relationship between farmers' income and governmental agricultural R&D expenditure, using concatenated statistical methods for time series evaluation. Statistical data for the period of 2004–2020 were retrieved from the Eurostat database. Because agriculture plays different roles in European Union member states' economies, we have considered it necessary to analyze the situations separately according to two criteria—the share of agriculture in GDP, and the time of country's accession to the EU. Thus, the countries with large shares of agriculture in the GDP, above the average of the European Union (1.63%), are Bulgaria, Cyprus, the Czech Republic, Spain, Estonia, Finland, France, Greece, Croatia, Hungary, Lithuania, Latvia, Poland, Portugal, Romania, the Slovak Republic, and Slovenia, as seen in Figure 1. The date refers to 2021. The second criterion, the time of accession of the country to the EU, divided them into old and new member states, considering 2004 as the threshold.

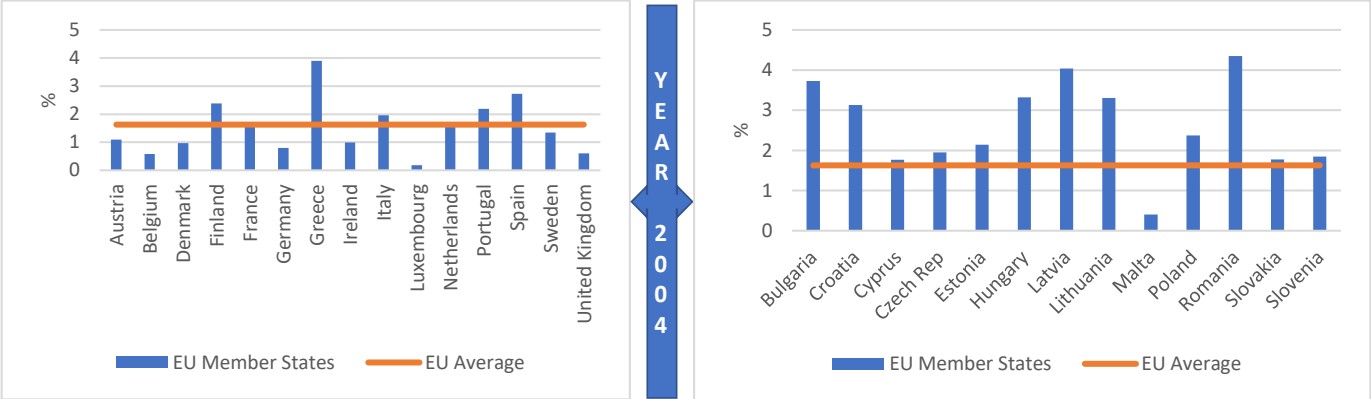

**Figure 1.** Year 2021 share of agriculture, forestry, and fishing value added in the GDP of EU28 member states (**left**: old member states; **right**: new member states). Source: edited by the authors from The World Bank [9].

The contribution of this paper to the literature is two-fold. First, two essential indicators of the Second Sustainable Development Goal (SDG 2), Zero Hunger, have been studied together—governmental agricultural R&D expenditure and farmers' income. Furthermore, several types of farmers' income indexes are used to measure sustainable farmers' income, considering that the results can faithfully reflect how agricultural R&D expenditure affect farmers' income. Secondly, the paper investigates farmers' income, within the context that ensuring a fair standard of living for farmers and contributing to the stability of their incomes are essential objectives for the Common Agricultural Policy of the European Union [10].

The paper is structured with five parts; following the introduction, the literature review describes the state of research in the fields of sustainable agriculture, sustainable development goals and income targets, agricultural investments, and research and development expenditure, in part two. The dataset regarding the R&D expenditure in agriculture and the farmers' income are analyzed, using the regression model, in part three, and the results are then discussed in part four. Finally, conclusions are drawn in part five.

## 2. Literature Review

The topic of sustainable agriculture is even more current as the recent trends in food prices and the unstable political, social, and economic conditions raise concerns about agricultural market equilibrium, food security, and farmers' income stability. Researchers [11] argue that agricultural R&D is a significant determinant of agricultural production and productivity and as a result, food prices and poverty.

Sustainable development of agriculture represents one decisive goal for the near future, and most states have adopted this approach in policy definition. The term, in essence, has diverse meanings, depending on the context, and also includes varying explanations and practices, such as farming methods and ecological stability. The concept is also known for aggregating economic aspects with resource conservation, maintenance, and improvement, concentrating on both the environmental and the ecological aspects. Depending on the reporting context, the sustainable agriculture focus can also vary, from yield improvement, crop diversity, and income prosperity to environmental stresses.

Gherardelli [12] claimed that one of the major challenges for governments is to ensure sufficient food for the population, taking into account the global emergencies of population and income growth, changes in diets, and decreasing availability of natural resources. These challenges call for increasing agricultural production, but in the context of a more economically, socially, and environmentally sustainable agriculture. Promoting a significant expansion of agricultural R&D and its funding could address this challenge.

Furthermore, trade-offs between agricultural productivity and sustainability have started to be studied. FAO reports [13] show that a major challenge for agriculture is to acknowledge and explore the potential trade-offs and contradictions between sustainability, with its environmental and social dimensions, and productivity, as its economic dimension.

The Second Sustainable Development Goal (SDG 2), Zero Hunger, is part of the UN's 17 Goals aimed at transforming the current world. They represent a ready-for-action guideline for governments all around the world, no matter the economical profile, that ensure sustainable three-layered growth (social, economic, and environmental). SDG 2 or the Zero Hunger ambition, is a comprehensive strategy with tangible goals and targets that tackle hunger elimination, food and nutrition security accomplishment, and agricultural sustainable development [14].

Particularly, our research will draw attention to the importance of R&D investment in promoting a sustainable agricultural development in countries with various agri-profiles; specifically, the concerned SDG 2 targets consist of targets 2.3—agricultural incomes and 2.A—increased investment in agricultural research. The first one, agricultural incomes, is relevant to be studied because supporting farmers' incomes and stabilizing them remain essential objectives of the Common Agricultural Policy of the European Union, as stated by the European Commission [10]. As declared by the UN [14], SDG target 2.3 aims to double the agricultural productivity and revenues of small-scale food producers, family farmers, etc., including secure and equal access to resources and inputs, knowledge, and financial services. The second goal, SDG target 2.A, aims to increase investment in rural infrastructure, agricultural research and extension services, technology development, etc. [14]. The linkages between the two targets are explored in the current research, assuming that investments in agricultural R&D are one of the key drivers of farmers' income growth, as suggested by FAO [15].

Investments in SDG 2, specifically through continuous worldwide focus on parents receiving more socio-economic attention in order to provide the needed food for their children, as well as on smallholder farmer empowerment and agricultural sustainable development, gender equity in farming and socio-farming, institutional de-formalization, have to be overseen as a holistic SDG approach that definitely boost collateral targets and objectives [16].

Research literature highlights how to intervene regarding the global food security topic; however, effective actions, strategies, and assessment methods remain challenging.

Authors summarize some aspects that could improve general food security achievement through targeted agricultural interventions in food security, measurable actions that address food security, and improved systematic methodological reviews as methods for agricultural interventions [17].

Scientific papers also suggest leveraging know-how into concrete action plans towards reducing SDGs disparities through research, industrial, political, and consumer collaboration; there is also an acute need of implementation [18]. Another important aspect besides implementation is the threat-constrain identification in applying existing knowledge, as the incapacity to deliver could further increase the SDG gaps.

Evidence from the research literature also points towards an existing causal relationship between agricultural output and domestic agricultural investment [19], whereas the current paper emphasizes the governmental agricultural R&D investment as a possible defining pillar of sustainable rural economic growth.

Other studies have reviewed the foreign non-governmental investment in agriculture and several short-run and long-run effects have been noted, with the prompt recommendation of improving the absorption capacity and administrative fluency [20].

Some studies have discussed the controversial effects of the agricultural technology investment on farmers' income. On the one hand, researchers found that the agricultural R&D had a positive effect on the growth of farmers' income [21]. The authors of [22] explored the possible ways to double the farmers' income and concluded that this objective can be achieved if the stakeholders follow a comprehensive and targeted approach regarding income opportunities, including investment in agricultural R&D and infrastructure. It has been found [23] that, when the agricultural modernization starts, per capita income increases. On the other hand, some studies show that the agricultural investments have limited positive effect on farmers' income [24].

Income and, as such, the economic growth, enter into discussions about sustainability. The authors of [25] claimed that economic growth is sustainable only if it is compatible with environmental quality. This argument is controversial: some researchers argue that the economic growth is deteriorating the environment [26], while others [27] state that the link between income growth and environmental degradation is insignificant.

Governmental agricultural research plays an essential role in the development of modern agricultural scientific breakthroughs, high quality economical results, and productivity increases [28]. Generally, according to economic theory [29], R&D expenditure is considered to be essential for economic growth, development, and sustainability. The agricultural R&D expenditure have positive effects on famers' functional distribution and scale distribution of income, while agricultural technology promotion expense has negative effects. In comparison, the first relationship mentioned has a stronger link than the second [7]. The authors of [30], in their analysis of the impact of R&D on productivity, claimed that R&D positively drives productivity.

The basic economic theory provides few ideas about R&D expenditure. Thus, researchers tried to fill in this gap with findings regarding either the relationship between agricultural output and R&D expenditure or the R&D as a source of efficiency [31,32], or the long-run relationship between productivity, as a dependent variable, and R&D expenditure, as an independent variable [33].

Evidence from the research literature also proved that R&D capital investment is mandatory to determine productivity amplification in the agri-food industry; while in the study of [34], it is also stated that the R&D expenditure should improve at a faster pace than the output's, as a direct result of the transferability drawback in the technological sector of agriculture across countries or businesses, and also due to the fact that conservative research levels should be prevented from plunging.

There is a large body of empirical studies on the research and development expenditure in agriculture. When interrogating the Web of Science (WoS) database using the following parameters: "agricultural R&D" and "agricultural income", 295 results were reported, most studies being written in the following domains: environmental science

(21%), economics (15%), environmental studies (12%), etc. The researchers' interest in the way the agricultural R&D expenditure impacted the farmers' income increased after 2010, when, on average, 10 articles were published in the journals indexed in WoS, reaching a peak of 48 in 2021. Using the VoS Viewer Software to see the linkages between agricultural R&D and income and other related topics, five clusters were identified. They comprise themes such as yield, management, performance, efficiency, quality, systems, soil, sustainability, emissions, economic growth, agricultural productivity, food security, trade, consumption, income, poverty, policy, and strategies, demonstrating the interest of researchers in these topics in connection with agricultural R&D investments.

Regarding the relationship between agricultural R&D expenditure and income, the authors of [35] argued that agricultural R&D has made important contributions to reducing poverty in South Asia in the post Green Revolution period. The authors of [3] found that public agricultural R&D has a positive impact on rural household income. It was found in [7] that the agricultural R&D expenditure has positive effects on famers' functional distribution and the scale distribution of income, while agricultural technology promotion expenditure has negative effects. In comparison, the impact of the former is larger than that of the latter. Some authors [36] explored the R&D expenditure for new technology in livestock farming and argued that investments in research and development lead to more efficient and sustainable resource management for developing countries.

Innovation-led economic growth has proved to gain popularity among governments, a process that is also known as smart growth [37], which is growth that is also applicable in the agricultural sector, therefore highlighting the importance of R&D as a main driver of innovation.

Income in agricultural holdings has been studied in the context of the sustainable development of agriculture [38], the authors exploring the inequalities among farms and demonstrating that the process of the concentration of land and capital led to an increase in farmers' income disparities. Farmers' income has been explored in relation to rural tourism [39], and it was found that rural tourism positively and significantly affects sustainable farmers' revenues, although, among different types of farmers' income, the magnitude varies.

Based on the analysis of previous studies, the question is raised regarding how governmental agricultural R&D expenditure affect farmers' revenues, expecting a direct and intensive relationship between them. Thus, a hypothesis is put forward as follows:

**Hypothesis 1 (H1).** *Governmental agricultural R&D expenditure influences farmers' income, and the extent varies between countries with different agricultural profiles.*

The research objective is to investigate the relationship between governmental agricultural R&D expenditure and farmers' revenues, with the final purpose of better managing the paths and directions of governmental agricultural R&D investments towards achieving the sustainable development goals, including farmers' income growth.

## 3. Materials and Methods

Statistical data regarding the government support of agricultural research and development in the EU member states are presented in Figure 2. Significant structural differences between countries can be noticed across multiple states.

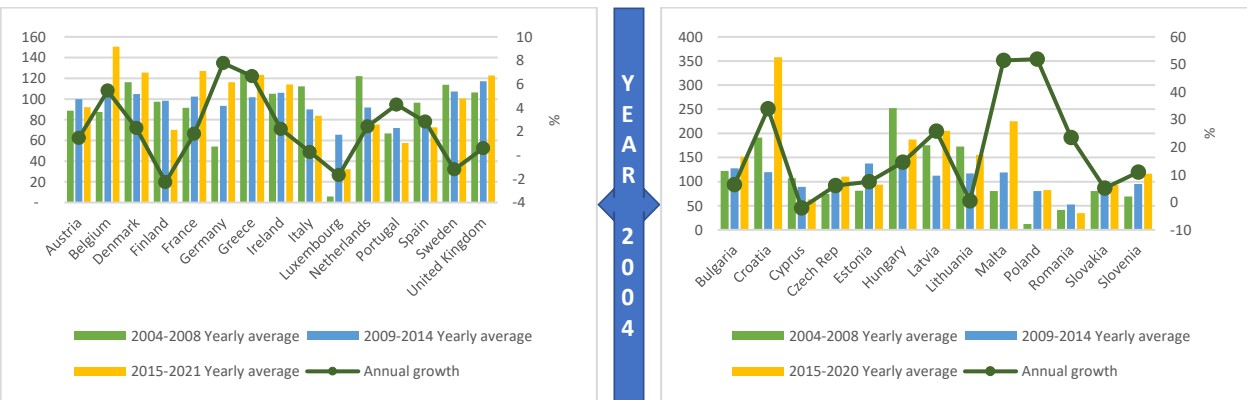

**Figure 2.** Annual average for government support of agricultural research and development (**left** axis) (Variable R&D in the model) and the average annual growth (**right** axis) at the EU28 member state level, in percentages (**left**: old member states; **right**: new member states). Source: edited by the authors from Eurostat [40].

The government support of agricultural research and development, representing the independent variable (R&D) in the current paper's tested model, is part of the European Union Sustainable Development goals, targeting SDG 2, Zero Hunger. It is referring to the governmental allocation of the budget for each member state for research and development activities in the agricultural sector, signaling how prioritized the public funding of research and development is for each state.

The visual representation of the dependent variable tested in the econometric model, Governmental Support to Agricultural R&D (variable R&D), is illustrated in Figure 2, which concatenates the yearly average for the mentioned interval on the left hand axis and the year over year average growth on the right hand axis. Both representations are illustrated in percentages. It aims to describe the most representative financing period for the research side in agriculture and signals in which member state this expenditure is gaining popularity. In the case of yearly growth, there are a few states that raise attention, as their values surpass 20% of annual growth, such as Poland (52%), Croatia (34%), Latvia (26%), and Romania (24%), and the remaining countries account for less than 10% of the yearly increase. Meanwhile, Cyprus, Finland, Luxembourg, and Sweden recorded negative values.

As the variable is described through indexes, the three-colored chart would best describe which period recognized R&D as being essential. What can be easily observed is that either period 2004–2008 or period 2015–2020 are more visible for most of the states. As the country comparison could not be performed for the annual average, due to the reporting method of the dataset, and as each year's value for a given member state is reported for the year 2010, the average annual increase was compared.

The most representative sectors of the economy where R&D has been financed are clearly indicated in Figure 3. Western European countries set the trends, but also the orientation of the investment profile. The largest investment in R&D has been recorded in the German R&D industry. In the agricultural sector, the countries with the largest investments are Germany (USD 1883 million), Spain (USD 1510 million), and the United Kingdom (USD 1016 million), for the old EU member states, and for new member states, the largest investments in agricultural R&D are recorded in Poland (USD 261 million), the Czech Republic (USD 181 million), and Hungary (USD 162 million). For the rest of the sectors for both old and new countries, only the largest investment will be presented: environment R&D in Germany (USD 1908 million) and Poland (USD 318 million); energy R&D in Germany (USD 2989 million) and the Czech Republic (USD 157 million); defense R&D in France (USD 5297 million) and Poland (USD 335 million); education R&D in Germany (USD 947 million) and Poland (USD 302); and industry R&D in Germany (USD 8397 million) and the Czech Republic (528 million).

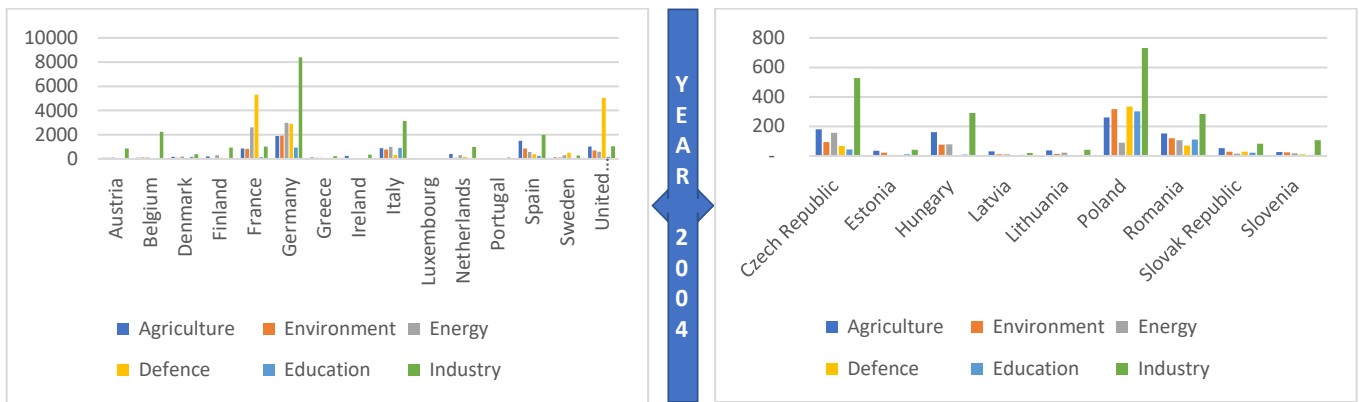

**Figure 3.** Cumulative R&D governmental expenditure by economic sector for the 2005–2020 period (in million USD, 2015 constant) (**left**: old member states; **right**: new member states). Source: edited by the authors based on available data [41].

As top R&D investments have already been discussed in Figure 3, Figure 4 presents the R&D expenditure share by sector for each individual member state. Industrial R&D was expected to represent a significant part of total governmental R&D for the developed countries. The largest agricultural R&D shares were recorded in Ireland (33.5%), Spain (27%), and The Netherlands (20.4%), for old member states, while for new member states, Latvia has recorded the largest agricultural R&D share with 38.2%, followed by Lithuania (29.7%), and Estonia (28.2%).

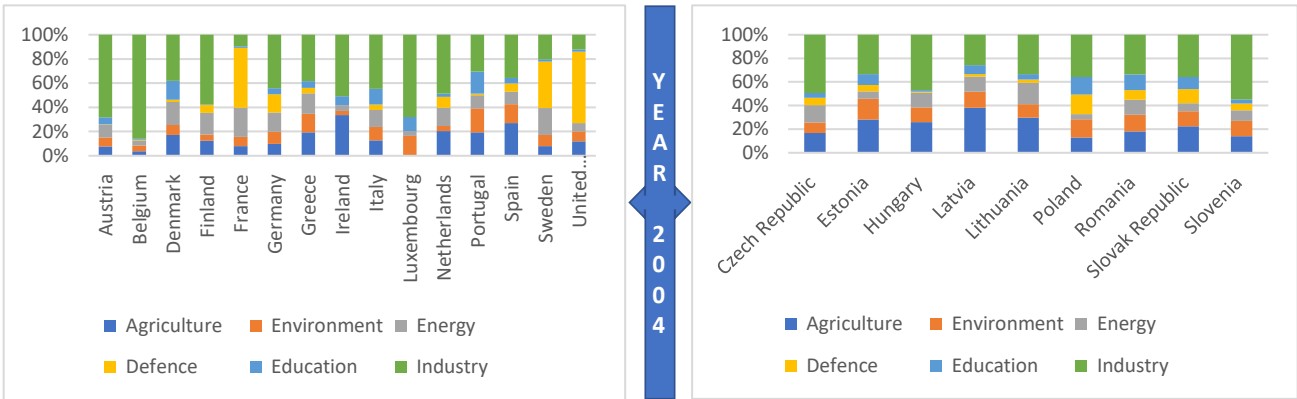

**Figure 4.** Share of cumulative R&D governmental expenditure by economic sector in percentages (**left**: old member states; **right**: new member states). Source: edited by the authors based on available data [41].

The dependent variable of the model tested in this particular research, entitled real income of factors in agriculture per annual work unit (INC), is also part of the European Union Sustainable Development Goals indicator set. It aims to monitor the SDG progress towards reaching the Zero Hunger ambition. The indicator has an additional scope as part of the Common Agricultural Policy objectives, to be more precise. The described indicator is an aggregation of income factors generated by agricultural activities, such as remunerated factors of production—capital, wages, and land, either owned, borrowed or rented, according to Eurostat, the issuing entity—and it represents a description of all factors of production (inputs, depreciation, taxes, and subsidies).

A visual representation of the independent variable tested in the econometric model, real income of factors in agriculture (variable INC), is illustrated in Figure 5, which concatenates the yearly average for the mentioned interval on the left hand axis and the year over year average growth on the right hand axis. Both representations are reflected in percentages, aiming to describe the most representative financing period for the agricultural income; therefore, the annual average country comparison could not be performed

due to the reporting method of the dataset, as each year's value for a given member state is reported for the year 2010 against the average annual growth, which can be compared.

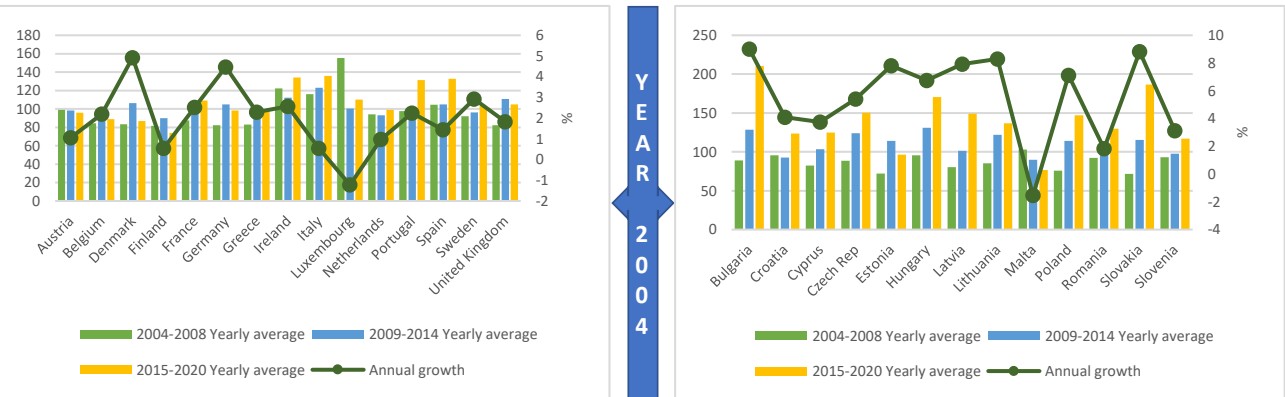

**Figure 5.** Annual average of real income of factors in agriculture per annual work unit (Variable INC in the model) (**left** axis) and average annual growth (**right** axis) at the EU28 member state level, in percentages (**left**: old member states; **right**: new member states). Source: edited by the authors from Eurostat [42].

Significant yearly growths, have been recorded for the following states: Bulgaria and Slovakia (9%), Estonia, Latvia, and Lithuania (8%), and Hungary and Poland (7%); while on the opposite side, besides Luxembourg and Malta's negative growth, Austria, Finland, Italy, The Netherlands, and Spain all recorded just 1% annual growth. For the 2004–2008 period, Luxembourg and Ireland recorded the largest annual average for the agricultural income, with Slovakia the lowest average, while for the period 2009–2014, Hungary reported the highest value, and for the period 2015–2020, Bulgaria recorded the largest growth.

The EU28 territory is explored in this particular research, as the EU countries are known to possess either developed or developing status, and it has already been demonstrated how important R&D investment is for sustainable development. The potential for agricultural-rural socio-economic development lies in the governmental investment profile, especially expenses that concentrate on bringing novelty.

As referring to the research methodology, the scientific literature indicates similar statistical approach when farmers' incomes are assessed; the R-squared concept, together with linear regression, have been used to determine the influence level of several socio-economic variables regarding the agricultural cooperative income [43]. Other research papers applied similar econometric approaches to draw the interdependence between agricultural product purchasing power and investment in the agricultural and processing sector, along with other independent variables, to demonstrate the importance that agriculture plays in sustainable economic development [44].

In order to be able to compute the statistical data as extracted from Eurostat, multiple transformation procedures were required, especially for the independent variable (R&D), as originally it has been described in absolute values (as seen in Figure 6). Therefore, a duplication was mandatory for the indexing methodology of INC, determining annual figures to be reported up to the year 2010.

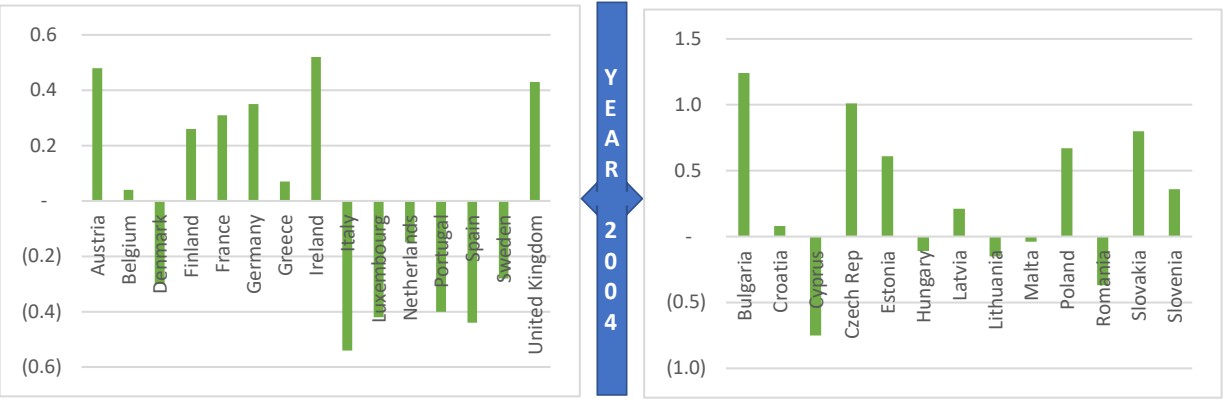

**Figure 6.** Coefficient of the independent variable (governmental agricultural R&D expenditure) in the regression equation (**left**: old member states; **right**: new member states). Source: edited by the authors.

## 4. Results and Discussions

The regression model has been performed individually, across all the EU28 member states, for the dependent variable, farmers' income, and the independent variable, agricultural R&D expenditure. Using statistical software to perform the analysis for each country, the resulting econometric R-squared values are graphically represented in Figure 7 for each individual country, together with the independent variable coefficient, as seen in Figure 6.

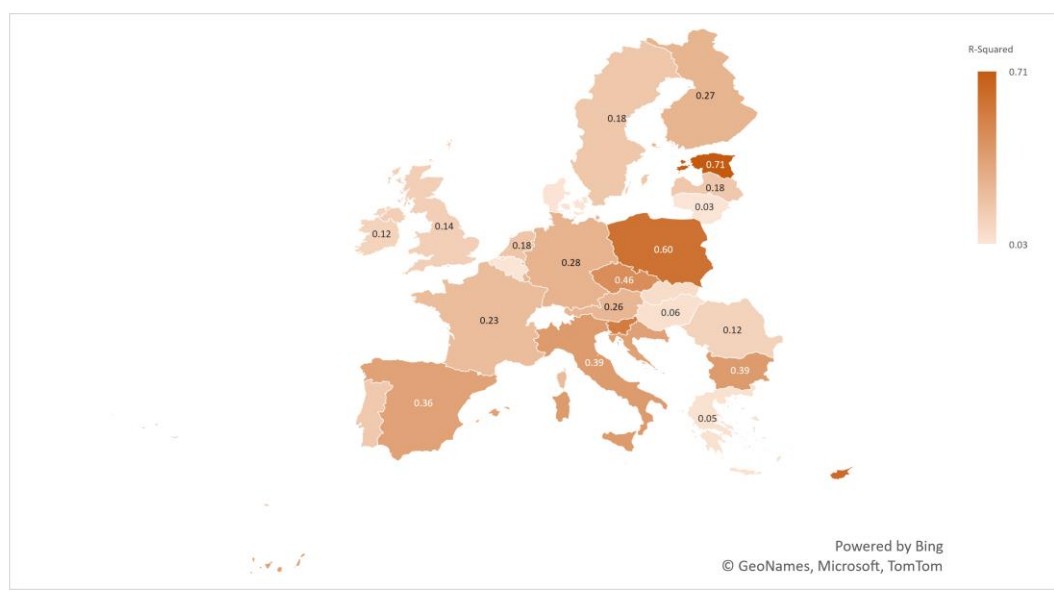

**Figure 7.** R-squared values for the EU28 member states of the agricultural governmental R&D (R&D) expenditure's capacity for explaining the real income of factors in agriculture per annual work unit. Source: edited by the authors.

Top performing R-squared parameters in the present model have been registered in the countries of Estonia (0.71), Cyprus (0.63), Poland (0.60), Slovenia (0.54), and the Czech Republic (0.46), while the lowest performing parameters have been recorded in Belgium and Latvia (0.03), Denmark (0.04), Greece (0.05), Hungary (0.06), and Slovakia (0.07). Thus, for the first group of countries, the levels of governmental agricultural R&D expenditure strongly influence the levels of farmers' income, and any change in their allocation may significantly change the farmers' well-being and rural economies' prosperity. For the second group of states, the levels of governmental agricultural R&D investments slightly influenced the levels of farmers' income. Each member state value reflects a specific national

profile, including the agricultural profile; therefore, drawing a one-sided direction line would be a matter of partial subjectivity. Moreover, other variables of the economic, natural, technological, social, or political kind, which are not presented in the current analysis, clearly impact the farmers' incomes.

$$INC = R\&D \times Coeff + C \tag{1}$$

Equation (1) R&D capability of explaining INC (using the coefficients stated in Figure 8, if the hypothesis is valid). Source: edited by the authors.

| Member State | R2 | X coeff |
|---|---|---|
| Austria | 0.26 | 0.48 |
| Belgium | 0.03 | 0.04 |
| Denmark | 0.04 | (0.30) |
| Finland | 0.27 | 0.26 |
| France | 0.23 | 0.31 |
| Germany | 0.28 | 0.35 |
| Greece | 0.05 | 0.07 |
| Ireland | 0.12 | 0.52 |
| Italy | 0.39 | (0.54) |
| Luxembourg | 0.26 | (0.42) |
| Netherlands | 0.18 | (0.15) |
| Portugal | 0.17 | (0.40) |
| Spain | 0.36 | (0.44) |
| Sweden | 0.18 | (0.28) |
| United Kingdom | 0.14 | 0.43 |

YEAR 2004

| Member State | R2 | X coeff |
|---|---|---|
| Bulgaria | 0.39 | 1.24 |
| Croatia | 0.37 | 0.08 |
| Cyprus | 0.63 | (0.75) |
| Czech Rep | 0.46 | 1.01 |
| Estonia | 0.71 | 0.61 |
| Hungary | 0.06 | (0.11) |
| Latvia | 0.18 | 0.21 |
| Lithuania | 0.03 | (0.15) |
| Malta | 0.09 | (0.04) |
| Poland | 0.60 | 0.67 |
| Romania | 0.12 | (0.37) |
| Slovakia | 0.07 | 0.80 |
| Slovenia | 0.54 | 0.36 |

**Figure 8.** Directions and intensities of the relationships between governmental agricultural R&D expenditure and farmers' income for each EU member states and the UK (**left**: old member states; **right**: new member states). Source: edited by the authors.

The values of $R^2$ and of the coefficient of governmental R&D expenditure in agriculture in relation to farmers' income is summarized in Figure 8. Four clusters have been identified:

(i)    Countries where governmental R&D expenditure in agriculture are among the key factors of farmers' income growth and positively impact them: Estonia, Poland, and Slovenia;

(ii)   Countries where governmental R&D expenditure in agriculture are among the key factors of farmers' income and negatively impact them: Cyprus;

(iii)  Countries where governmental R&D expenditure in agriculture are not considered among the key factors of farmers' income growth and positively impact them: Bulgaria, the Czech Republic, Finland, France, Greece, Croatia, Latvia, and Slovakia;

(iv)   Countries where R&D governmental expenditure in agriculture are not considered among the key factors of farmers' income growth and negatively impact them: Spain, Hungary, Lithuania, Portugal, and Romania.

In order to assess how a 1% increase in INC is determined by coefficient % increase in R&D and properly use Equation (1) for a member state, it is also important to take into consideration the C (constant) value that varies from −21.53 to 177.1, with an average value of 91.75.

The threshold value of $R^2$ from which R&D governmental expenditure in agriculture is considered to be among the key factors of farmers' income is 0.5. The countries that registered values of the share of agriculture in GDP lower than the average of the European Union, 1.63%, as seen in Figure 1, may be excluded from the results' analysis:

Austria, Belgium, Germany, Denmark, UK, Ireland, Italy, Luxembourg, Malta, The Netherlands, and Sweden.

For some member states, as seen in Figure 6, as the governmental R&D expenditure in agriculture increases, the resulted coefficients indicate a positive relationship. The positively impacted farmers' income due to R&D expenditure growth are registered in the following countries: Bulgaria, the Czech Republic, Finland, France, Greece, Croatia, Latvia, Poland, Slovakia, and Slovenia. Similar results have been found by other authors [38], who demonstrated that the budgetary support for agriculture reduced the polarization and the inequalities of farmers' income. As stated in FAO reports [15], evidence from many countries shows that governmental agricultural R&D, education, and access to information for farmers lead to income growth. These results are consistent with those of other studies [21,22], showing that the agricultural R&D expenditure has a positive effect on the growth of farmers' income. Generalizing, investments in rural areas, not only in agriculture, have a positive impact on per capita income, as argued in the research literature [23].

For other member states, as the governmental R&D expenditure in agriculture increases, the resulted coefficients indicate a negative relationship (Figure 6). The negatively impacted farmers' income due to R&D expenditure increase are recorded in the following countries: Cyprus, Spain, Portugal, Romania, Lithuania, and Hungary. Since only for Cyprus is the relationship between the variable strong ($R^2$ = 0.63), while for the rest of the countries, the relationships are medium to weak, we may argue that these results do not change the assumption established at the beginning of the research. Moreover, although controversial, these results are consistent with those found in the literature [24], showing that the agricultural investments have a limited positive effect on farmers' income.

In order to find possible explanations for which in six countries with large shares of agriculture in the GDP and for which the governmental agricultural R&D expenditure negatively impacts the farmers' income (Figures 1 and 8), the government support for agricultural research and development in each EU member state in absolute values are taken into account (Figure 9). A total of 22 EU member states have allocated less than EUR 100 million per year for the governmental agricultural R&D expenditure. In all six countries, except for Spain, the levels of government support for agricultural research and development are very low, less than EUR 50 million. Future research should investigate the reasons why, in the case of Spain, the farmers' income is negatively impacted by the agricultural R&D, although in this member state, the government support is significant, at EUR 488 million.

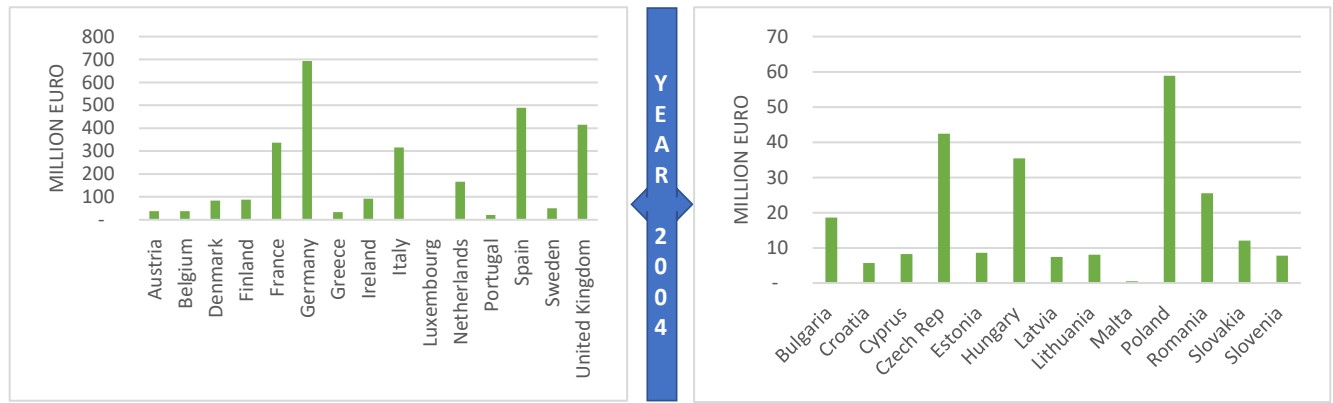

**Figure 9.** Government support for agricultural research and development by EU member state in absolute figures in millions of euros (annual average of 2004–2020 period) (**left**: old member states; **right**: new member states). Source: edited by the authors from Eurostat [40].

Forthcoming strategic directions may take into consideration improving the competitiveness of rural areas and creating new income and employment opportunities for

farmers and their families, as these remain major aims for the future of the European Union [45]. The results of this study have broader implications that go beyond the economic prosperity of farmers and rural areas towards those involving social issues. Income growth affects the farmer's willingness to remain in agriculture and to continue to produce food, while a decrease in income would create negative pressures on social welfare, and migration from rural to urban conditions [46]. As a result of this situation, production amounts will decrease, food prices will increase, food security will be jeopardized, and pressure on governmental financing will increase. Strategies and macroeconomic policies should consider not only the economic results, but also the social and ecological consequences, and the trade-offs and contradictions between sustainability, with its environmental and social dimensions, and productivity, as its economic dimension, should be acknowledged and explored.

## 5. Conclusions

This study presents the impact of the governmental agricultural research and development expenditure on farmers' income, in the sustainable development context. Statistical data were analyzed, using the regression model, for the time period of 2004–2020 for each member state of the European Union.

The results of the model are diverse, emphasizing the diversity of the EU's economy itself. For some countries, a significant portion of farmers' income growth is explained by governmental R&D expenditure in agriculture, e.g., Estonia, Poland, and Slovenia; for others, the farmers' incomes are partially explained by R&D investments in agriculture, e.g., Austria, Belgium, Bulgaria, the Czech Republic, Germany, Finland, France, Greece, Croatia, Ireland, Latvia, and Slovakia. Controversial results have been found for countries where the governmental R&D expenditure in agriculture negatively impacts the farmers' income. However, in the case of these countries, the farmers' income is weakly impacted by R&D expenditure, as the values of $R^2$ show. Thus, the hypothesis (H1): Governmental agricultural R&D expenditure influences farmers' income, and the extent varies between countries with different agricultural profiles, has been validated.

Two main categories of results were obtained—countries where farmers' income are influenced by the governmental agricultural R&D expenditure, and countries where the influence is weaker. For the first category of states, the governmental expenditure should continue to focus on ensuring sustainable income for farmers. The implications go beyond improving farmers' revenues, generating on- and off-farm employment, and contributing to strengthening the economic prosperity of the European Union rural communities. For the second group of countries, where the influence of agricultural R&D investments is weaker, the governmental expenditure may be directed towards strengthening rural development, promoting food quality, meeting safety standards and food security, and fostering animal welfare.

Considering this variety of results, their implications are also diverse. The findings should change the methods and directions for using the governmental agricultural R&D expenditure; for example, in countries where the R&D investments in agriculture are among the key factors of farmers' income growth, the governmental expenditure should be carefully underpinned by economic analysis. As such, the macroeconomic policy in rural areas and agriculture become effective in its pathway to achieving the sustainable development indicators.

Bearing in mind the controversial results of the research, in conclusion, in order to achieve the Sustainable Development Goals, Goal 2, Zero Hunger, and to ensure a fair standard of living for farmers and the stability of their incomes, as declared by the objectives of the EU's Common Agricultural Policy, a specific role should be attributed differently to agricultural policy instruments, in general, and governmental agricultural R&D expenditure, in particular, in each member state of the European Union.

The study has its limitations, since it takes into consideration only one of the farmers' income drivers, the governmental agricultural R&D expenditure. Future research and in-

depth analysis are required, which should include other factors influencing the agricultural income, such as weather conditions, market prices, factor productivity, production costs, supply chain fluency, economic and social crises, etc.

**Author Contributions:** Conceptualization, M.S., V.C.T., and R.A.I.; methodology, V.C.T.; software, V.C.T.; validation, V.C.T., M.S., and R.A.I.; formal analysis, I.C.N. and C.G.Z.; investigation, C.G.Z.; resources, V.C.T. and I.C.N..; data curation, V.C.T.; writing—original draft preparation, V.C.T.; writing—review and editing, R.A.I.; visualization, I.C.N.; supervision, M.S.; project administration, M.S.; funding acquisition, M.S. All authors have read and agreed to the published version of the manuscript.

**Funding:** This research received no external funding.

**Institutional Review Board Statement:** Not applicable.

**Informed Consent Statement:** Not applicable.

**Data Availability Statement:** The data presented in this study are available on request from the corresponding author.

**Conflicts of Interest:** The authors declare no conflict of interest.

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
