# Peer review of "The Influence of Governmental Agricultural R&D Expenditure on Farmers’ Income—Disparities between EU Member States"

_sustainability, doi:10.3390/su141710596_

Round 1

Reviewer 1 Report

I previously reviewed and commented on this manuscript when it was submitted to Sustainability. Now I see that the authors have revised the article, and they have taken into account all of my comments. My sole comment regarding the new submission is that the figures are of poor quality. The authors are kindly requested to redraw the figures in higher quality and resolution.
Best regards,

Author Response

Dear reviewer,

We are thankful for your detailed comments and the value that they brought to the article. This new submission has figures with higher resolution, both in text and in the attachment.

Best wishes.

Reviewer 2 Report

The authors have correctly corrected the comments I made in the previous review or logically explained the doubts I raised in the review.

Author Response

Dear reviewer,

We are thankful for your kind commentaries that brought increasing value to the paper.

Best wishes.

Reviewer 3 Report

Dear authors

thank you for taking into account my suggestions to expand some parts of the article. I think it is now ready for publication. Congratulations.

Author Response

Dear reviewer,

We are very thankful to your suggestions and glad to guide us.

Best wishes.

This manuscript is a resubmission of an earlier submission. The following is a list of the peer review reports and author responses from that submission.

Round 1

Reviewer 1 Report

The manuscript entitled " Agricultural R&D - a driver for sustainable agriculture by income growth " has been reviewed. This paper aims to confirm the claim that investment in agricultural research and development is one of the major determinants of farmers' income as one of the indices of sustainable development. The subject of the manuscript is very interesting, it is well written and contains important results that help the decision-makers. Some general and minor comments are provided below.

Title: The authors have investigated the governmental "Research and Development" (R&D) investments on farmers' income. As the non-governmental R & D contribution is not involved in the study, it would be expected to mention that in the title clearly. Also, as the direct link between the farmers’ income and sustainable development is not studied, So, the title could be revised in a way with more focus on what has been studied by the authors in the present manuscript.

Please add the titles of the vertical axes in figures 1, 2, 3, 4, and 6.

Please remove the background from all figures to make them more environmentally friendly when printed on paper.

While the authors have argued that the way that they used to present the results "would best describe", It's difficult for me to pick out the details, especially in figure 2.To help readers understand the findings, the authors are respectfully asked to modify the presentation or add further explanations to the text.

Page 5 Figure 2: The outcomes of different countries differ significantly.I propose that the authors divide the figure into two parts: one with countries with a 2000 or higher value (six countries) and the other with countries with less than a 2000 value.

Why is the classification of the examined time not consistent throughout the text? For example, a four-year period is studied for "Government support to agricultural research and development and the average annual growth at EU-28 Member State level" (Figure 1), while it is divided into 7-year periods for “Cumulative R & D governmental expenditure by economy sector” (Figure 2). Please revise or add more explanations.

What do the brown dots in figure 4 (annual growth) represent? It seems that they are not in their proper location. Please check.

Page 8, Table 1: Without providing the overall model, the authors have detailed the validation of the regression models for several countries. What are the model coefficients for each country? Can a single model be applied to all countries?

 Conclusion: The conclusion typically aims to summarize the study's results and discuss its limitations. Every study has its limitations, which the authors should state plainly without hesitation. The limitations of the study should be made evident at the end of the conclusion section since this study is not an exception.

Reviewer 2 Report

The entire article has a lot of pages, but nothing innovative is written in it. There is no in-depth statistical analysis. Statistical analysis was not sufficiently discussed. The entire manuscript looks like an excerpt from a report from some major work. Without statistics and research, the article does not contribute to any knowledge development. An expansion of literature, conclusions and discussions is required.

The discussion was not conducted in accordance with the guidelines of the journal. An expansion of literature, conclusions and discussions is required.

English needs a lot of improvement.

Reviewer 3 Report

he manuscript is very interesting, but the authors made only a random assessment of the factor affecting the income of farmers in EU countries.

My main comments are:

Explaining the increase in income through an increase in agricultural research expenditures without examining other factors affecting farmers' income is, in my opinion, an oversimplification. 

As is known in traditional agriculture especially crop production is largely determined by weather conditions which affects farmers' income. Income is affected by production costs and economic crises such as the one in 2008. 

The scientific purpose of the article is not presented. This objective should be presented at the end of the literature analysis.

In the M&M section, the authors practically did not present how the variables were calculated. No formulas were presented. The 4 graphs presented are not very clear and understandable. no information on what unit is on the vertical axis (except the one with %).

The presentation of incomplete data causes confusion in understanding the manuscript (figures 2 and 3 have fewer countries analyzed than figures 1 and 4).

From the methodology, it is difficult to understand what the authors want to analyze and with what tools. 

The results and discussion section presents interesting data, but in my opinion comparing economies with different economic potential is wrong.  

You can't compare state expenditures on research in the agricultural sector between Germany and Malta, for example. In my opinion, it would be necessary to divide these outlays by, for example, the value of GDP from agriculture in a country. 

Also of great concern to me is comparing the income of farmers in different countries without specifying whether all entities involved in agricultural production were analyzed?

Are EU subsidies added to farmers' income?

Reviewer 4 Report

Dear authors
Eurostat statistics show that in 2018, rural areas accounted for 83% of the total EU area and agricultural, forestry and natural areas accounted for 80% of the total EU area. However, countries have different agricultural potential, so should  countries such as the northern countries be excluded from the analysis or treated separately?
Also, it might be worthwhile to do a separate analysis for old and new EU members. I would argue that for new agricultural countries like Poland, investment in research in the agricultural sector is more important than in industrialised countries like Germany. How can you present sustainable rural development without a deeper analysis that looks at different indicators and causal factors. Your study is based on the assumption that investment in agricultural research and development (R&D) is one of the main drivers of farmers' income, but that in some countries agriculture is a priority and in others it is not, if only because of climatic conditions. The role of communities and rural areas in the economies of the former is steadily increasing, and discussions of macroeconomic policy may have focused on separate analysis of "agricultural" and "developed" countries when it comes to financing, investment, and public spending and the effectiveness of their implementation. In Poland, for example, investments contribute to the economic well-being of rural communities, and macroeconomic policies are considered effective.
When is agriculture sustainable? Is it sustainable when it provides a steady stream of agricultural products (thus meeting social and economic expectations) or when it is environmentally friendly?
 How does research impact the Sustainable Development Goal (SDG 2), Zero Hunger, which is part of the 17 goals from UN to transform the world today?
How can we intervene in the issue of global food security? What actions, strategies, and assessment methods remain a challenge?